# Green Template-Mediated Synthesis of Biowaste Nano-Hydroxyapatite: A Systematic Literature Review

**DOI:** 10.3390/molecules27175586

**Published:** 2022-08-30

**Authors:** Ferli Septi Irwansyah, Atiek Rostika Noviyanti, Diana Rakhmawaty Eddy, Risdiana Risdiana

**Affiliations:** 1Department of Chemistry, Universitas Padjadjaran, Jl. Raya Bandung-Sumedang KM 21, Jatinangor, Sumedang 45363, Indonesia; 2Department of Chemistry Education, UIN Sunan Gunung Djati Bandung, Jl. A.H. Nasution No. 105, Bandung 40614, Indonesia; 3Department of Physics, Universitas Padjadjaran, Jl. Raya Bandung-Sumedang KM 21, Jatinangor, Sumedang 45363, Indonesia

**Keywords:** hydroxyapatite, green synthesis, template, biowaste

## Abstract

Hydroxyapatite (HA) is a well-known calcium phosphate ingredient comparable to human bone tissue. HA has exciting applications in many fields, especially biomedical applications, such as drug delivery, osteogenesis, and dental implants. Unfortunately, hydroxyapatite-based nanomaterials are synthesized by conventional methods using reagents that are not environmentally friendly and are expensive. Therefore, extensive efforts have been made to establish a simple, efficient, and green method to form nano-hydroxyapatite (NHA) biofunctional materials with significant biocompatibility, bioactivity, and mechanical strength. Several types of biowaste have proven to be a source of calcium in forming HA, including using chicken eggshells, fish bones, and beef bones. This systematic literature review discusses the possibility of replacing synthetic chemical reagents, synthetic pathways, and toxic capping agents with a green template to synthesize NHA. This review also shed insight on the simple green manufacture of NHA with controlled shape and size.

## 1. Introduction

HA exists in nature and has long been recognized as an important material used to make dental fillings because of its excellent biocompatibility [1,2,3]. Biomedical uses include mammalian bone regeneration (osteoclast and osteoblast cells), bioimaging, cancer therapy, and drug loading [4,5,6]. Aside from biological uses, this ceramic substance has also benefited catalysis. When combined with other metals or metal oxides, the HA nanomaterial has been identified as an excellent material for catalytic applications [7]. For the many uses of HA, the crystal size of HA must always be scaled down to the nanoscale. NHA is readily explained because materials with smaller sizes (crystallite size) are more reactive and have better physicochemical qualities due to the significantly larger exposed surface area [2,3,4,5,6,7,8,9,10]. Recent research trends have focused on altering synthesis pathways to keep materials at the nanoscale due to the benefits of nanomaterial applications and their economic significance [11]. HA synthesized from chemical sources is complicated or biologically unsafe [12]. HA can be synthesized using chemical precursors, such as calcium and phosphorus [13]. Synthesizing HA from chemical sources in the form of phosphoric acid (H_3_PO_4_) and calcium hydroxide (Ca(OH)_2_) has been achieved [14]. HA from natural sources contains ions, such as Na^+^, Zn^2+^, Mg^2+^, K^+^, Si^2+^, Ba^2+^, F^−^ and CO_3_^2−^ [15]. Currently, many researchers are researching eggshells as an alternative to bone damage in humans because the eggshell contains a relatively high element of calcium carbonate (CaCO_3_), which is helpful as a biomaterial [16,17]. Moreover, eggshells are economical and easy to provide [18].

The method of the synthesis of nanoparticles widely uses natural material sources, such as bacteria, fungi, and plants, including mediated plant extracts offering environmentally friendly production techniques. In addition, the character of its products is varied and has several applications in different disciplines [19]. In addition to its environmentally friendly properties, natural sources act as the reducers and stabilizers of products and control their shape and size of nanoparticles. Natural sources are more effective in reducing and stabilizing than artificial precursors [20]. Therefore, exploring safer and environmentally friendly alternatives to NHA synthesis using natural sources as templates is essential. The most popular synthesis methods for nanoparticles are those mediated by plant extracts. Plant extracts are an alternative to common chemicals used to avoid agglomeration. Plant extracts are known to contain phytochemicals with anti-inflammatory and antioxidant properties. Therefore, it can regulate the shape and overcome the agglomeration barrier during the manufacture of nanoparticles [18,20].

The NHA synthesis process greatly determines the morphology, crystallography, and phase purity of the hydroxyapatite particles, which ultimately determines the mechanical properties of materials for biomedical applications [21,22,23]. The green synthesis process uses widely available plant extracts in large quantities, is safe to handle, non-toxic, and highly effective under various temperature, pH, and salinity conditions. In addition, using plant extracts in the green synthesis process is a widely accepted and relatively more efficient technology for preparing NHA as a biomedical application [24]. Several recent studies have shown that the synthesis of NHA was successfully carried out using naturally available banana peel waste as a template [25,26,27,28]. Inside the banana peel, pectin is also extant, the amount of which varies. The pectin content in banana peels is about 1.92 to 3.25% of the dry weight [27]. It was found that pectin is rich in carboxyl and hydroxyl groups. Pectin can stimulate the binding of calcium ions (Ca^2+^) from the solution to carboxylic ions, which initiate the nucleation and growth of hydroxyapatite crystals [28]. Pectin is a polysaccharide material with biocompatible, biodegradable, and critical biological properties, such as antimicrobial, anticoagulant, and anti-inflammatory properties. Pectin has shown potential in biomedical applications for bone tissue regeneration [29].

A few reviews focus on obtaining hydroxyapatite from natural sources [5,20,29,30,31]. However, no work has yet been published that focuses on the concurrent discussion of the biowaste sources of hydroxyapatite and its green template-mediated synthesis. Therefore, this paper focuses on the literature on the recent investigations of the synthesis of NHA preparation with a reliable green template and materials approach. In addition, this review article also describes the sustainable synthesis of NHA in various types of biowaste.

## 2. Methodology

A systematic literature review (SLR) is utilized here, which is a literature review that discovers, assesses, and interprets all data on a study issue to answer previously specified research questions [32]. In December 2021, the literature search was restricted to items published between 2012 and 2022. The title and keywords “green synthesis hydroxyapatite” and “hydroxyapatite for biomedical application” and “Nano-Hydroxyapatite” AND “Template” AND “Banana” OR “Musa paradisiaca” AND “Biomedical” were used to search for publications in research databases at Sciencedirect, Pubmed, and Researchgate.

The Preferred Reporting Item for Systematic Reviews and Meta-Analytic (PRISMA) technique was utilized. All publications that passed the selection procedure were examined and summarized based on the objectives, year of publication, document type, publication stage, keywords, and source type. The inclusion criteria are (1) studies on green synthesis of hydroxyapatite with banana (Musa paradisiaca) as a template for biomedical applications and (2) research articles published in peer-reviewed journals. The exclusion criteria include (1) a general study performed on biomaterial for biomedical applications and (2) articles containing a literature review or a meta-analysis. The search begins by analyzing the titles and abstracts of all search results and comparing them to predefined criteria.

## 3. Result

The research database search resulted in all keywords search results obtained 257 research articles in total, from Sciencedirect, 232 articles; Pubmed, 12 articles; and MDPI, 13 articles. After scanning the title, the same article was in three different databases. The results after deducting the duplicates are 232 articles. A total of 160 results were excluded because they were in the form of an article review (30 articles), publication time did not meet the criteria (130 articles), or they did not meet the topic criteria. Still, they were concerned with other applications of hydroxyapatite (31 articles) or biomedical applications from a different biomaterial (30 articles). There are 17 articles included in the final literature review. The literature search is described in more detail in Figure 1.

### Green Synthesis of NHA

Green synthesis uses ecologically friendly ingredients in nanoparticles, such as bacteria, fungi, and plants. Plant extract-mediated synthesis offers an environmentally friendly production technique and has several applications in various research disciplines [19]. Green sources reduce and stabilize agents to produce shape- and size-regulated nanoparticles. The synthesis of nanoparticles utilizing contemporary techniques has emerged as a significant application in the biomedical and human healthcare fields for many goods. Nanotechnology, in general, is described as the manipulation of materials at the atomic level by a mix of technical, chemical, and biological processes [33]. Creating green nanoparticles is a procedure that combines plant biotechnology with nanotechnology. Plant and fruit extracts are commonly employed to create semiconductor and metal nanoparticles. Natural sources are more effective in reducing and stabilizing than manufactured precursors. Plant extracts contain metabolites, such as sugars, polyphenols, terpenoids, alkaloids, proteins, and phenolics, reducing metal ions into nanoparticles and resulting in particle strengthening. Plants have been extensively studied as a source for transforming inorganic metal ions into nanoparticles. Plant extract proteins and metabolites play an important role in metal ion reduction. Using plant extracts in nanoparticle synthesis includes less hazardous waste creation, lower maintenance and waste disposal costs, extract function as both a reducing and stabilizing agent, and good impacts on therapeutics [19,34]. Figure 2 shows that the primary benefit of nanoparticle biosynthesis over conventional techniques is its environmental friendliness and low cost.

The fundamental technique (as shown in Figure 3) for preparing nanoparticles includes collecting the plant or a portion of the plant, followed by thorough washing. The cleaned plant components are dried for several days before pulverizing using blenders. The dry granules are cooked in distilled water to make the final infusion, filtered, and used to make metal nanoparticles [35]. Larger nanoparticles are often generated at pH levels ranging from two to four. The reported size of nanoparticles at lower pH ranges from 20 to 80 nm. Small particles are generated at higher or alkaline pH [36]. Using tansy fruit extract, described the influence of temperature [37]. The reaction rate increases as the temperature rises, further boosting nanoparticle formation [24]. Due to the Instability of nanoparticles, an optimal period is necessary for complete nucleation and subsequent nanoparticle stability [38,39]. 

Hydroxyapatite can be synthesized in a variety of ways (methods). Some methods are solid-state reactions, microemulsions, hydrothermal, sol-gel, and template-guiding [40]. There are three classifications in the HA synthesis method—the dry method, the wet method, and the high temperature method. This can be seen in Figure 4.

The agglomeration problem affects all of the processes outlined above. Aggregation or agglomeration occurs during the synthesis of various nanomaterials and the production of HA. Because of the interfacial area’s limits, agglomeration decreases nanoparticles’ mechanical characteristics [40,41,42,43]. The agglomeration problem is overcome by introducing capping agents or surfactants to ensure excellent particle dispersion by preventing the attraction interactions between the nanoparticles [44,45]. Another reason to prefer HA produced using easily accessible natural precursors is the material’s high cost. Traditional HA preparation necessitates using high-purity commercial reagents, which are costly [46,47,48,49,50].

There is always a need to reduce the size of HA crystals to the nanoscale in diverse HA applications. This is because materials with a smaller size (crystal size) are more reactive and increase physicochemical properties because the surface area exposed is larger. This positions the synthesized material with nano-sizes preferred as a better option. Recent research trends also point to synthesis route engineering to keep materials on a nanoscale [3].

HA obtained from natural resources or biological waste, shells, cow bones, cuttlefish bones, fish bones, and camel bones has shown better metabolic activity and increased bioactivity than other synthetic products. An additional advantage of naturally derived HA is the presence of carbonate and citrate groups and the low potassium, sodium, magnesium, and strontium content in their chemical composition. Another essential factor to consider is the calcium/phosphorus ratio. In synthetically prepared apatite, this ratio is usually lower than natural HA. Table 1 shows several research results of HA synthesis using natural Ca sources from animal bone waste using different synthesis methods to produce different Ca/P ratios.

Some of the advantages of HA synthesis using natural materials, especially biowaste can be seen in Table 2.

## 4. Discussion 

### Green Template Mediated Synthesis of NHA

Plant extract is a green alternative to standard chemicals used to avoid agglomeration. In addition, plant extracts include phytochemicals with anti-inflammatory and antioxidant properties. As a result, they are employed to regulate the shape and overcome the agglomeration barrier while creating a broad spectrum of nanoparticles [19,20,21]. Figure 5 shows the procedure for forming NHA using a green template for biomedical applications (bacterial activity parameters and toxicity test).

Hydrothermal synthesis is a reasonably simple, intrinsically scalable approach and significantly more chemically friendly than many other nano-production technologies [58]. Hydrothermal synthesis is distinguished by high repeatability and control over microstructure [59]. The surface characteristics of nanoparticles may be significantly changed from hydrophilic to hydrophobic by hydrothermal synthesis by selecting the right surface coating agent [24]. Surface modification has improved the dispersion properties of inorganic nanoparticles in an organic polymeric matrix [60]. NHA particles were developed in various shapes, including nanoplates and nanorods. Surface characteristics of NHA nanoplates were modified using several organic surfactants, including polymeric and dicarboxylic ligands. Surface treatment using a polymeric surfactant resulted in a complete shift in surface characteristics from hydrophilic to hydrophobic. This method may improve the compatibility of inorganic nanoparticles and organic biopolymers. Surface modification with a di-carboxylic ligand can also provide exfoliated or delaminated HA plates into the host organic medium [60].

Figure 6 shows the mechanism for the synthesis of NHA nanoparticles. As seen in the mechanism, pectin biomolecules include a carboxyl group that serves as a reactive group. The electrostatic interaction of the carboxyl group (COO^−^) found in the polymeric backbone of the pectin with Ca^2+^ ions result in the production and nucleation of the NHA (Ca^2+^-COO^−^) complex. This process is known as calcium immobilization. The functional groups on the surface of polymers play an essential role in nucleation due to their varying efficiency in grabbing calcium ions. Due to the effects of supersaturation, there was an ionic interaction of PO_4_^2−^ ions to the complex and nucleated the NHA-pectin composite with the addition of phosphate solution to the calcium pectinate (Ca^2+^-COO^−^) complex. The removal of pectin moieties occurred at higher temperature calcination, resulting in the synthesis of NHA nanoparticles. As a result of this process, one may conclude that pectin functions as an effective chelating agent in the synthesis and nucleation of NHA.

The template addition approach is frequently used to monitor the shape and size of product nanoparticles via the development of ordered micelle surfaces. Many scholars have addressed the role of an additional organic template in the nucleation of HA ceramic nanoparticles [61]. The approach is most typically used to supplement other synthesis procedures (for example, hydrothermal, co-precipitation, and ultrasonic) as an effective and simple method to create nanoparticles with regulated shape and low agglomeration [7]. A wide range of synthetic macromolecular organic compounds (cationic [62] and anionic surfactants [63], block copolymers [64], double hydrophilic block copolymers [65], and polyvinyl alcohol [66]) have been applied as surfactant templates to synthesize HA nanorods or needles [67]. 

Biosurfactants are helpful as environmentally acceptable alternatives to synthetic surfactants. As a result, it is critical to investigate safer and more environmentally acceptable options for the biosynthesis of HA nanoparticles utilizing natural sources as templates or surfactants. The plant extract-mediated green approach is now the most prevalent nanoparticle manufacturing technique [21]. Plant extracts are widely accessible in massive amounts, safe to handle, non-toxic, and highly efficient under high temperatures, pH, and salinity conditions. Furthermore, using plant extracts is a well-acknowledged approach for manufacturing nanoparticles that are easily scalable and less costly [68]. Plant extracts from grape seed, Moringa Oleifera leaf and flower, tamarind extract, Gum-Arabic, calendula flower, papaya leaf, banana, potato and orange peel, *Azadirachta indica* and *Coccinia grandis* leaf extract, mango, and others are used as reducing, stabilizing, or capping agents [20,64]. 

Biological materials have been shown to offer a high potential for green synthesizing beneficial and vital compounds, such as HA nanoparticles. In addition, they are safe, plentiful, widely available, completely regenerable, non-exotic, inexpensive, and capable of supporting fast development [6,47]. *Parkia biglobosa*, sometimes known as the African locust bean, is a significant leguminous tree found throughout Africa, South America, and Asia. It is a plentiful economic tree in Nigeria, uniformly distributed throughout. The sweet yellowish powder buried in the economic seed of *P. biglobosa* is employed to prepare locust beans [6].

Hydroxyapatite nanoparticles were prepared using pectin from *P. biglobosa* pulp as a template. The hydroxyapatite nanoparticles’ antibacterial properties were investigated using suitable analytical methods. As a result, hydroxyapatite generated at low pectin concentrations is suitable for application in biomedical sciences. HA produced with 0.1 percent pectin appears somewhat porous, distinct, and flaky, with less aggregation. In terms of shape and size, the particles appear to be non-uniform. The HA’s shape at 0.5 percent pectin seemed to be spherical and agglomerated compared to HA produced at 0.1 and 1 percent pectin. This shows that pectin acts as a template for HA synthesis and determines the type of particles generated. Polysaccharides have been shown to alter the physical characteristics of nanomaterials when used as templates [67,68,69]. 

Low crystalline HA is required in biological sciences due to its great resorbability in vivo [70]. At low pectin concentrations (0.1 percent), discrete nano HA particles with low crystallinity and good purity are generated. Compared to HA nanoparticles produced in the absence of pectin, it is clear that HA nanoparticles generated in the presence of 1 g of pectin demonstrated higher antibacterial activity as seen by the large zone of inhibition. The characteristics of HA nanoparticles are always connected to their shape and size, which are heavily controlled by a variety of synthesis parameters, such as time, pH, temperature, precursor concentration, and the presence and type of organic templates [66,71]. 

Licorice (*Glycyrrhiza glabra*) with glycyrrhizic acid (GA) and some polyphenolics as its primary water-soluble triterpenoid saponin ingredients; licorice root extract (LE) is known to have anticancer, antiallergenic, antiviral, antidiabetic, antiulcer, anticonvulsant, antithrombic, antioxidant, and secretolytic and expectorant effects [72,73,74]. Among the other ingredients in LE, the polyphenolic compounds (including isoliquiritigenin, liquiritigenin, isoliquiritin, and liquiritin) have a high affinity to metal ions due to the many phenolic hydroxyls. In addition, plant polyphenolics have been recently applied as chelating or stabilizing agents to synthesize inorganic materials [75,76,77,78,79]. 

In the current investigation, HA nanoparticles were synthesized following a green route using licorice (*Glycyrrhiza glabra*) root extract (LE) as a natural green template [24]. The results suggest that the green synthesis of HA nanoparticles in LE through a microwave-assisted hydrothermal route has led to nanoparticles of more uniform shape and consistency in size nanoparticles than the control sample synthesized in the absence of the LE extract. This difference may be related to favoring the template role of the environmentally friendly (green) LE in the early seeding stage and the eventual formation of highly crystalline HA [76]. The main ingredient of LE is glycyrrhizic acid (GA). GA molecules formed nanorod micelles in the starting solution mixture of LE and calcium salt. The surfaces of the micelles are homogeneously incorporated with the polyphenolics (such as isoliquiritigenin). Ca^2+^ ions then formed Ca-polyphenolic complexes by interaction with the OH groups of the polyphenolics at the micelles’ surfaces (by a p-track conjugation effect). On subsequent addition of phosphate solution, PO_4_^3−^ groups are reacted with the Ca^2+^ side of the complexes to form HA–polyphenolic complexes on the nanorod-shaped patterns of the micelles [79]. The complexes are finally decomposed, and the HA precipitate is ripened into a nanorod HA product by the hydrothermal treatment. 

Hydroxyapatite (HA) nanorods were successfully synthesized in the presence of licorice root extract (LE) as a natural green template through microwave hydrothermal synthesis without any toxic chemicals. Crystalline HA samples were successfully synthesized in a slightly acidic medium at a relatively low temperature through a short time of hydrothermal processing. Compared to a control sample of HA synthesized without LE, the formation of uniform, well-defined shape, highly crystalline hexagonal HA nanorods in LE was confirmed. This study validates a novel eco-friendly green synthesis of HA nanorods through a synthetic surfactant-free rapid route. The synthesis is based on licorice, *Glycyrrhiza glabra*, and root extract as a template. The product HA nanorods can be widely applied in many biomedical fields, such as bone repair, drug delivery, and restorative dentistry [24]. Furthermore, polyphenolic OH groups have a high affinity for metal ions. The plant extract is commonly used as a reducing, stabilizing, and chelating agent [76,77]. 

*Moringa oleifera flower* (MOF) is also known as horse radish and drumstick. MOF has high antioxidant properties because of the abundance of polyphenols, alkaloids, tannins, flavonoids, vitamins, minerals, and carotenoids. MOF is extensively used in green synthesis and works as a stabilizing agent because of its high tannin content. Hence it was chosen for this investigation. As a result, a dry cake was crushed to yield Hydroxyapatite (HA) powders, and MOF extract capped Hydroxyapatite (MOFE: HA). According to the findings of this work, MOF extract Hydroxyapatite nanorods MOFE: HA can be green produced swiftly utilizing MOF extract. These nanorods are affordable, non-toxic, and environmentally benign, with an average size of 41 nm and rod shapes. Greenly manufactured MOFE: HA nanorods were more effective against Gram-positive bacteria than Gram-negative bacteria. MOFE: HA nanorods have also shown antifungal efficacy against common pathogenic fungi. The findings of this study demonstrated a wide variety of significant prospective uses of MOFE: HA nanorods in biomedical sectors.

Due to the general use of organic agents during the reduction process, green synthesis pathways based on plant extracts [78,79] provide an excellent option to address this issue. Templates enhance the biocompatibility and bioactivity of the resulting nanoparticles when plant-derived materials are used [80,81]. HA is no exception; as a result, the danger of toxicity is low [81,82]. Excess or unreacted plant extracts can be washed out. Even if they end up in the environment, they disintegrate quickly due to environmental variables. Plant extracts have an advantage over their synthetic counterparts as templates for biomedical HA synthesis. Table 2 shows several studies on NHA synthesis using green templates from plant waste. The results showed differences in the shape and size of the NHA. Therefore, the different sources of green templates affect the particle size and shape of the NHA.

The plant extracts performed experimentally provide various characterizations that lead to identifying compounds with nano-sizes of different shapes. For example, there are those in the form of spheres, rods, cubic, and triangles. In addition, plant extracts have often been used to produce nanoparticles by modifying the size, shape, and morphology of the surface, which plays an essential role in controlling the physical and chemical form of the nanoparticles [82]. Green templates for natural precursors that prevent agglomeration are often called plant extracts. Some of the green templates in the NHA Synthesis show different effects on the size and shape of the nanoparticles. This can be seen in Table 3.

Plant extracts are known to contain phytochemicals that have reduced and antioxidant effects. Therefore, it is often used to control morphology and overcome agglomerations while preparing various nanoparticle syntheses. Several reviews focus on the process of obtaining hydroxyapatite from natural sources. Most of the previous research also provided information on natural calcium sources. Discussing the importance of phosphate precursors from natural sources, such as plants is rare. The discussion of hydroxyapatite preparation approaches using materials of plant origin (biomolecules), which usually includes chelating agents, surfactants, templates, ligands, structure-directing agents, the use of phosphate sources from nature, obtaining phosphate sources from vegetable waste, and the synthesis of hydroxyapatites directly from plants without the use of conventional calcium or phosphate precursor materials [19,78].

Green synthesis, especially that which utilizes plant extracts, is not only for synthesizing nano-HA alone but for various nanoparticle syntheses. Some of these nanoparticles are even applied in the biomedical and pharmaceutical fields. Therefore, the use of plant or plant extracts for synthesis should be observed. A potential solution incorporating extraction techniques is using less solvent volumes, which are less time-consuming, cheaper, and can be optimized to extract phytochemicals from plants [85].

## 5. Conclusions

This article describes current plant components as reducing or stabilizing agents in manufacturing nanostructured hydroxyapatite. The advantages of employing a green template with traditional reducing agents have been examined. Making NHA and other calcium phosphate nanoparticles from plants have also been mentioned. This review has further highlighted the ecologically beneficial aspect of NHA synthesis using plant-mediated techniques. Green templates can modify the size, shape, and morphology of surface, which is essential for influencing the physical and chemical form of the nanoparticles. However, future research should involve several clinical trials for more specific biomedical applications to implement the resulting NHA in this field.

## Figures and Tables

**Figure 1 molecules-27-05586-f001:**
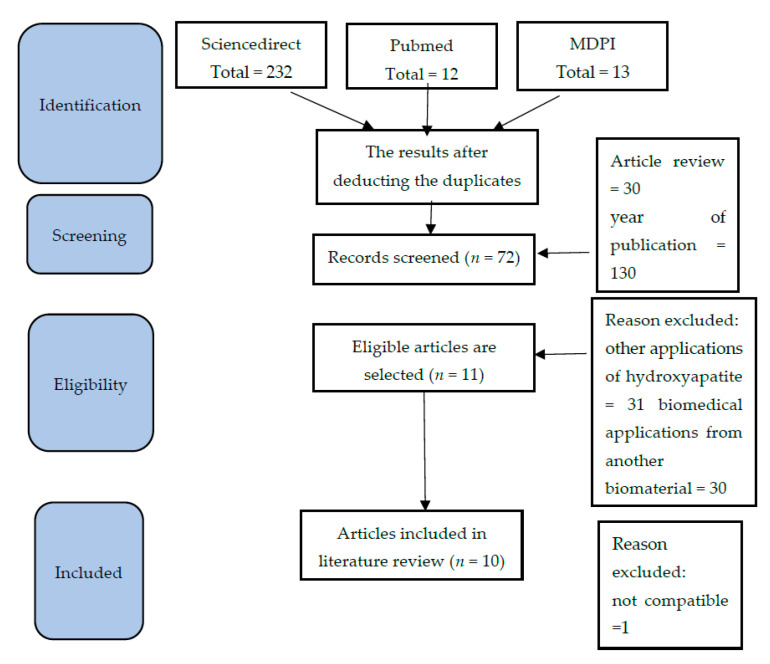
Literature Review Search Method.

**Figure 2 molecules-27-05586-f002:**
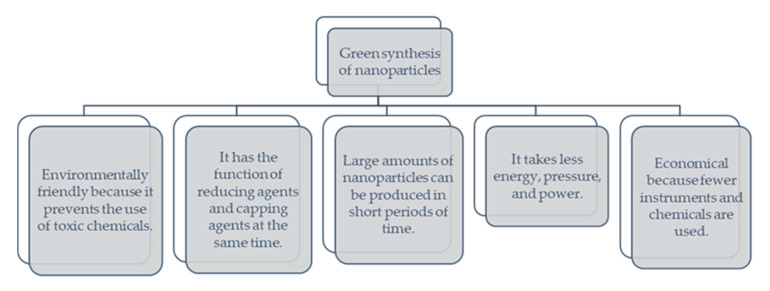
Advantages of green synthesis of nanoparticles, adopted from [20].

**Figure 3 molecules-27-05586-f003:**
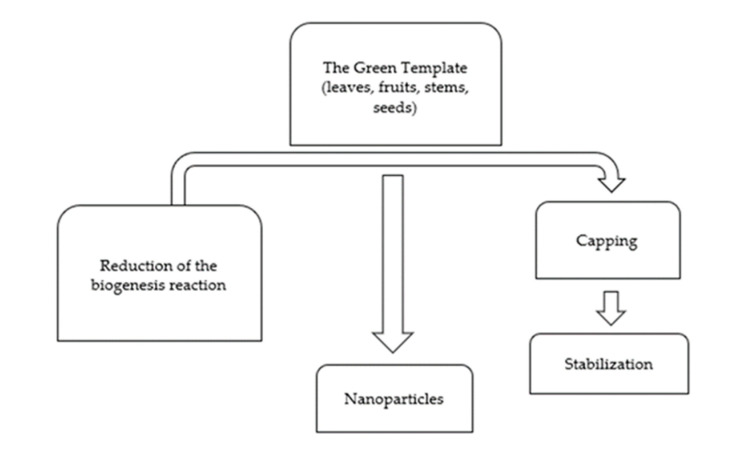
The basic process of the green synthesis of nanoparticles, adopted from [20].

**Figure 4 molecules-27-05586-f004:**
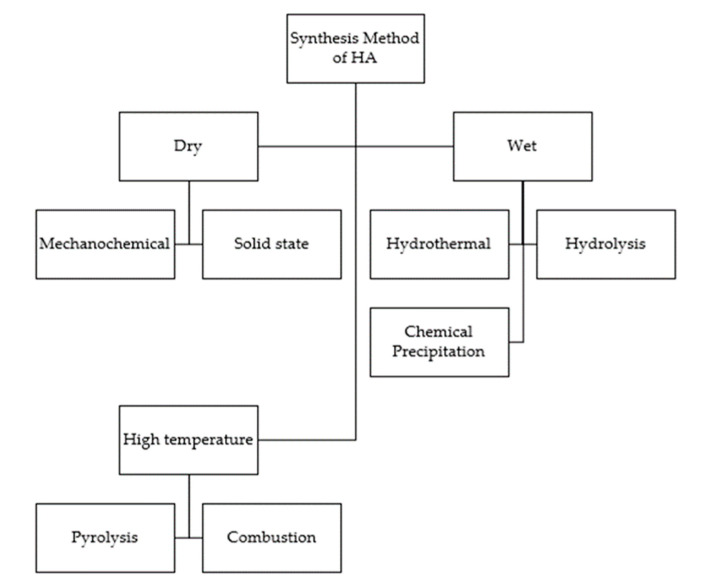
Classification synthesis method of HA, adopted from [40].

**Figure 5 molecules-27-05586-f005:**
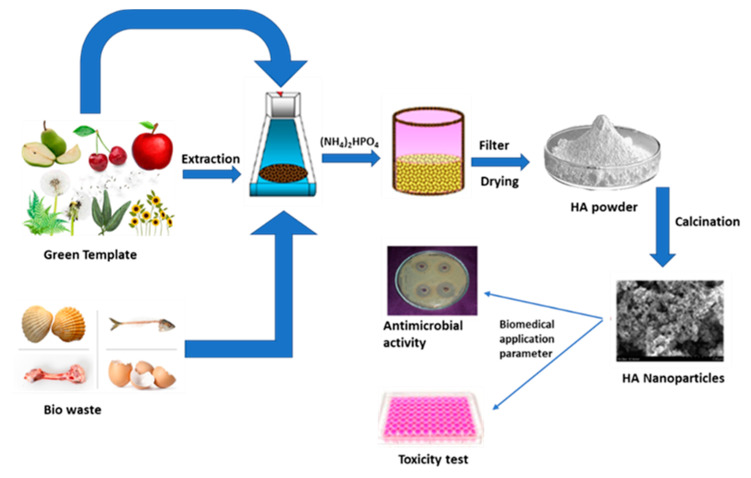
The procedure of formation of HA nanoparticles with a green template adapted from [24,25].

**Figure 6 molecules-27-05586-f006:**
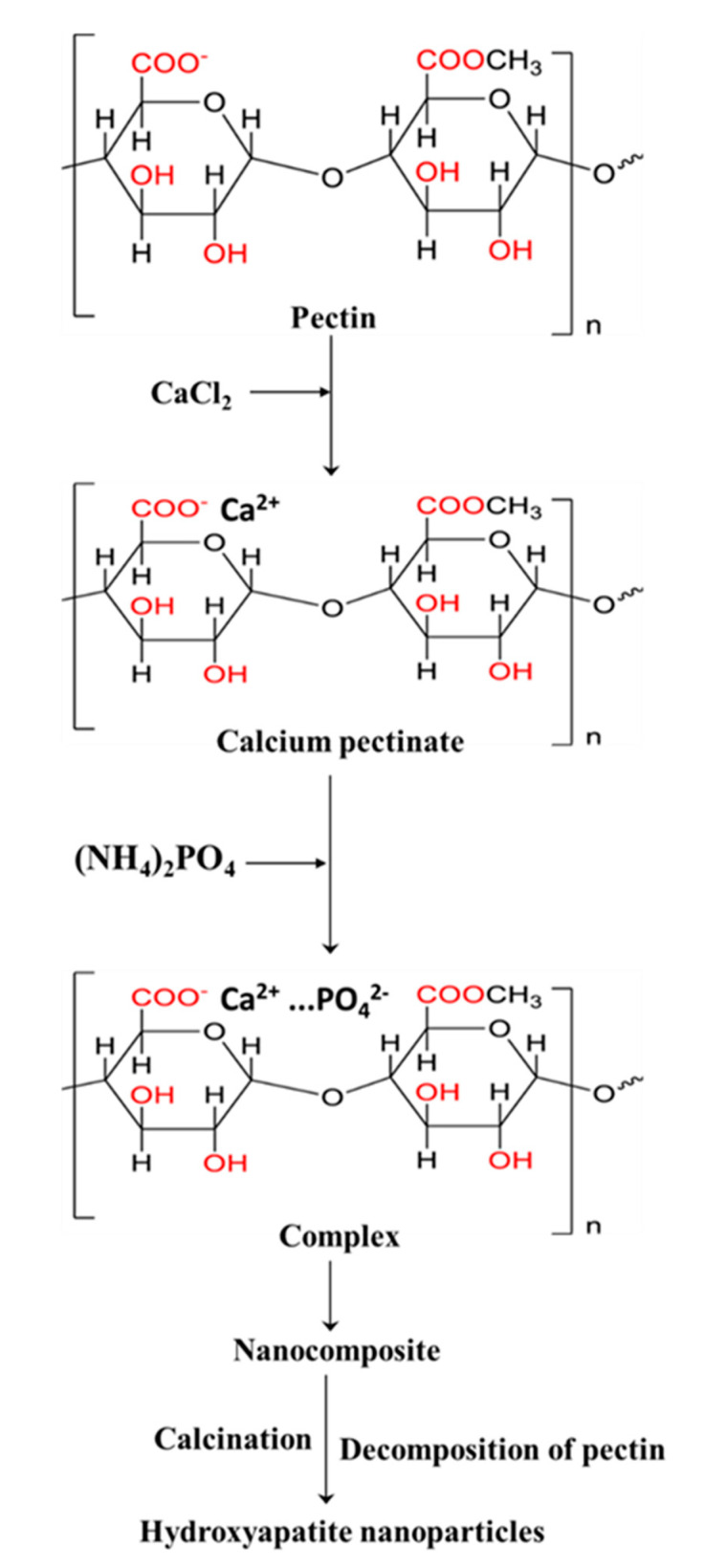
The mechanism for forming HA nanoparticles is adopted from [24,34].

**Table 1 molecules-27-05586-t001:** Results of various studies for the synthesis of Bio-waste HA.

Source	Synthesis Method	Ca/P	Ref.
Cow bone	Sintering	2.23–1.95	[47]
Camel’s bone	Calcination treatment	1.6557	[48]
Turkey thigh bone	Ball milling dan sintering	1.664	[49]
Fishbone	Co-precipitation	1.65	[50]
Cow, goat, and chicken bone	Sintering	1.57, 1.58 and 1.62	[51]
Chicken bone	Calcination	1.653	[52]
Fishbone	Hydrolysis and thermal process	1.69, 1.80, 1.82 and 1.83	[53]
Pork bone	Thermal process	1.64	[54]

**Table 2 molecules-27-05586-t002:** The advantages of HA synthesis using natural materials.

Advantages	Ref.
Produce relatively safe waste	[34]
Lower maintenance and waste disposal	[34]
Reducing agent	[20]
Stabilizing agent	[55,56]
Shape-controller and overcoming the agglomeration barrier	[19,21]
Increase mechanical strength	[25,26,57]

**Table 3 molecules-27-05586-t003:** Green Template on NHA Synthesis.

Source of Green Template	Size (nm)	Shape of Nanoparticles	Ref.
Pectin from Culinary Banana Bract	20–50	spherical	[28]
Pectin from Banana Peel	25–47	spherical	[25]
Licorice root	105	irregular sticks and perfect rods	[24]
Pectin from Parkia biglobosa pulp	17.5–26.3	non-uniform	[6]
Pectin from Opuntia ficus Indica peel	25	granular	[26]
Natural rubber latex	8	nanorod and nanoplate	[83]
Pectin from the citrus fruit peel	401.5–942.2	Sponge	[83]
Opuntia mucilage from opuntia ficus-indica cladodes	70–140	Spherical	[84]

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
