# Peer review of "Green Template-Mediated Synthesis of Biowaste Nano-Hydroxyapatite: A Systematic Literature Review"

_molecules, 2022, doi:10.3390/molecules27175586_

Round 1

Reviewer 1 Report

After reviewing the manuscript entitled " Green Template-Mediated Synthesis of Biowaste 2 Nano-Hydroxyapatite: A Systematic Literature Review" sent the following comments: The authors present an interesting topic.

1-      However, they must argue the advantages of using hydroxyapatite from natural sources compared to other materials in separate tables.

2-      Why is this research important?

3-      Show in the table the (advantage and dis advantage of different conventional methods) of synthesized of hydroxyapatite-based nanomaterials.

4-      In the abstract give brief information about some results of the synthesis of Bio-waste Nano-Hydroxyapatite.

5-      In the introduction; how does this study relate to a priori scientific studies and works?

6-      How do the design and hypotheses of the study relate to each other?

7-      Adjust the chemical structure in the whole manuscript (subscribe or subscript), for example, line 40.

8-      Mention in the figure; some applications of hydroxyapatite-based nanomaterials in different sectors.

9-      In the conclusion section state the added value of the main findings, and important discoveries explained in the manuscript and state some recommendations.

Author Response

Dear reviewers,

We want to thank you for some comments and suggestions from reviewers to our manuscript. Below are our responses to the reviewer’s comments and suggestions.

Reviewer’s 1 Comment:

After reviewing the manuscript entitled " Green Template-Mediated Synthesis of Biowaste 2 Nano-Hydroxyapatite: A Systematic Literature Review" sent the following comments: The authors present an interesting topic.

Respond:

Thank you for your valuable comment and suggestion. We revised some parts of our manuscript to increase the quality of our manuscript.

Reviewer’s Comment

Point 1: However, they must argue the advantages of using hydroxyapatite from natural sources compared to other materials in separate tables.

Respond 1: We added some data following the suggestion from the reviewer (lines 248-251)

Point 2: Why is this research important?

Respond 2: The topic in this research is very important to provide information on alternative pathways for synthesizing Hydroxyapatite Nanoparticles using natural waste materials with templates from plants or plants (lines 46-74). So, the results will contribute to environmental sustainability. In addition, the synthesis of NHA can be applied in several fields, including biomedical applications such as bone tissue replacement, dental implants, and drug delivery.

Point 3: Show in the table the (advantage and disadvantage of different conventional methods) of synthesized of hydroxyapatite-based nanomaterials.

Respond 3: If what is meant is the advantages and disadvantages of different synthesis methods of hydroxyapatite nanoparticles, then this is not the main focus of the discussion of this review article. The discussion of the hydroxyapatite synthesis method only mentions its categorization (lines 204 – 221).

Point 4: In the abstract give brief information about some results of the synthesis of Bio-waste Nano-Hydroxyapatite.

Respond 4: We modified some sentences in the abstract following the suggestion from the reviewer (lines 18-19).

Point 5: In the introduction; how does this study relate to a priori scientific studies and works?

Respond 5: We have added a discussion of previous studies on this topic and wrote about the importance of reviewing this study to get a comprehensive picture of alternative green synthetic pathways so that they can be developed into more practical research (lines 65-80).

Point 6: How do the design and hypotheses of the study relate to each other?

Respond 6: This manuscript belongs to the category of review article types, especially systematic literature reviews (SLR). In this article, the method section must explain how the methodology is to search for articles to be reviewed. In this article, the PRISMA method is used to search the article (line 89).

This is also in accordance with the criteria for SLR articles published in several journals, including Molecules (https://doi.org/10.3390/molecules26165057; https://doi.org/10.3390/molecules26123702; https://doi.org/10.3390/molecules26227062).

Therefore, the study design and its hypotheses will be related. By using the Systematic Literature Review, the resulting hypothesis indicates that NHA can be synthesized using the green synthesis pathway with the addition of templates from plants or plants.

Point 7: Adjust the chemical structure in the whole manuscript (subscribe or subscript), for example, line 40.

Respond 7: Revised

Point 8: Mention in the figure; some applications of hydroxyapatite-based nanomaterials in different sectors.

Respond 8: The topic in this review article is more focused on discussing green NHA synthesis as an alternative pathway for synthesis. Includes a discussion of the effects of adding a green template to the size and shape of NHA. Therefore, the discussion on applying NHA is not the main discussion. The application topic can be recommendations for another discussion in the article review because those topics are too broad. So, to fulfill the reviewer's suggestion in this section, we have modified the sentence.

Point 9: In the conclusion section state the added value of the main findings, and important discoveries explained in the manuscript and state some recommendations.

Respond 9: We modified some sentences in the conclusion following the suggestion from the reviewer. The main findings and important discoveries show in lines 495-496. Some recommendations show in lines 497-498.

Reviewer 2 Report

The review is a general study on the following issues: “This systematic literature review discusses the possibility of replacing synthetic chemical reagents, synthetic pathways, and toxic capping agents with a green template to synthesize NHA. This review also shed insight on the simple green manufacture of NHA with controlled shape, surface characteristics, mechanical strength, and other parameters that serve as criteria for biomedical applications.”

However, these issues have not been addressed. Synthetic chemical reagents, synthetic pathways, and toxic capping agents should be presented. Nanoparticle shape, surface characteristics, and mechanical strength have to be described.

The Methodology section should be developed to describe the methodologies of the achievement of HA nanoparticles. At present, this section describes the methodology of searching the articles for the review.

The results of experiments should be described in the Results section – it also needs to be developed.

Lines from 210 to 212 should be developed. The HA synthesis reactions should be shown on schemes. The methodology should be described.

Line 179 – what metal do the authors mean?

Author Response

Respond to Comments of Reviewers

Dear reviewers,

We want to thank you for some comments and suggestions from reviewers to our manuscript. Below are our responses to the reviewer’s comments and suggestions.

Reviewer’s 2 Comments:

The review is a general study on the following issues: “This systematic literature review discusses the possibility of replacing synthetic chemical reagents, synthetic pathways, and toxic capping agents with a green template to synthesize NHA. This review also shed insight on the simple green manufacture of NHA with controlled shape, surface characteristics, mechanical strength, and other parameters that serve as criteria for biomedical applications.”

Respond:

Thank you for your valuable comment and suggestion. We revised some parts of our manuscript to increase the quality of our manuscript.

Reviewer’s Comment

Point 1: However, these issues have not been addressed. Synthetic chemical reagents, synthetic pathways, and toxic capping agents should be presented. Nanoparticle shape, surface characteristics, and mechanical strength have to be described.

Respond 1: The topic in this review article is more focused on discussing green NHA synthesis as an alternative pathway for synthesis. Includes a discussion of the effects of adding a green template to the size and shape of NHA. Therefore, the discussion on synthetic chemical reagents, synthetic pathways, and toxic capping agents is not the main discussion. It also includes a discussion of surface characteristics and mechanical strength as recommendations for further discussion because these topics are characterizations for discussing NHA applications, especially in the biomedical field. So, to fulfill the reviewer's suggestion in this section, we have made some modifications to the sentence, especially in the abstract and conclusion.

Point 2: The Methodology section should be developed to describe the methodologies of the achievement of HA nanoparticles. At present, this section describes the methodology of searching the articles for the review.

Respond 2: This manuscript belongs to the category of review article types, especially systematic literature reviews. In this article, the method section must explain how the methodology is to search for articles to be reviewed. In this article, the PRISMA method is used to search the article (line 89).

This is also in accordance with the criteria for SLR articles published in several journals, including Molecules (https://doi.org/10.3390/molecules26165057; https://doi.org/10.3390/molecules26123702; https://doi.org/10.3390/molecules26227062).

Point 3: The results of experiments should be described in the Results section – it also needs to be developed.

Respond 3: As previously stated, this document falls under the category of article review. However, no findings from the experiment were discovered. What is written and discussed is a review and analysis of experimental data in earlier research relating to the review article's topic. For example, as stated in the discussion section (lines 385 - 451).

Point 4: Lines from 210 to 212 should be developed. The HA synthesis reactions should be shown on schemes. The methodology should be described.

Respond 4: Revised (Can be seen in lines 204 – 221)

Point 5: Line 179 – what metal do the authors mean?

Respond 5: The metals in question are several types of inorganic metal elements commonly used to form nanoparticles together with plant extracts. Among them are Gold (Au), Silver (Ag), Iron (Fe), Copper (Cu), and Zinc (Zn).
